# Validation of blue- and clear-native polyacrylamide gel electrophoresis protocols to characterize mitochondrial oxidative phosphorylation complexes

Jana Aref[1¤a], Seungtae Lee[1¤b], Supachaya Sriphoosanaphan[2,3,4], Micol Falabella[5], Shi-Yu Yang[1], Jan-Willem Taanman[1]*

**1** Department of Clinical and Movement Neurosciences, Queen Square Institute of Neurology, University College London, London, United Kingdom, **2** Institute for Liver and Digestive Health, University College London, London, United Kingdom, **3** Division of Gastroenterology, Department of Medicine, Faculty of Medicine, Chulalongkorn University, Bangkok, Thailand, **4** Centre of Excellence in Liver Diseases, King Chulalongkorn Memorial Hospital, Thai Red Cross Society, Bangkok, Thailand, **5** Department of Neuromuscular Diseases, UCL Queen Square Institute of Neurology, University College London, London, United Kingdom,

¤a Current address: Department of Neuromuscular Diseases, UCL Queen Square Institute of Neurology, University College London, London, United Kingdom
¤b Current address: Biodonostia Health Research Institute, San Sebastián, Spain
* j.taanman@ucl.ac.uk

## Abstract

The mitochondrial oxidative phosphorylation (OXPHOS) system plays a pivotal role in the cell's energy conversion. The enzymes involved in OXPHOS are arranged in five protein-lipid complexes. The first four complexes (I–IV) form the mitochondrial respiratory chain, while Complex V is an $F_1F_o$-ATP synthase. Mutations in genes involved in the biosynthesis of the OXPHOS complexes are an important cause of metabolic diseases. Blue-native polyacrylamide gel electrophoresis (BN-PAGE), originally developed by Hermann Schägger in the 1990s, has become instrumental in gaining insights into structure/function relationships of the OXPHOS system, including: (1) the assembly pathways of the complexes, (2) the composition of higher-order respiratory chain supercomplexes and (3) pathologic mechanisms in patients with a monogenetic OXPHOS disorder. We have used BN-PAGE for >20 years and validate here our recently published step-by-step laboratory protocol. This protocol describes the manual casting of native mini-gels and sample preparation for the resolution of individual OXPHOS complexes or respiratory chain supercomplexes. In addition to BN-PAGE, we explain the closely related clear-native (CN)-PAGE and two-dimensional BN/denaturing-PAGE techniques. Downstream applications include western blot analysis and in-gel enzyme activity staining for Complexes I, II, IV and V. Limitations of the technique are the comparative insensitivity of in-gel Complex IV activity staining and the lack of in-gel Complex III activity staining. Compared to other published BN-PAGE protocols, our protocol contains a shortened sample extraction

**Data availability statement:** All relevant data are within the manuscript and its Supporting information files.

**Funding:** JWT, Fund 42, Royal Free Charity, https://www.royalfreecharity.org, The funders had no role in study design, data collection and analysis, decision to publish, or preparation of the manuscript.

**Competing interests:** The authors have declared that no competing interests exist.

procedure, advises when to use BN-PAGE and when to use CN-PAGE, and suggests a simple enhancement step for in-gel Complex V activity staining that markedly improves sensitivity. Our protocol is adaptable and yields robust, semi-quantitative and reproducible results.

## Introduction

The oxidative phosphorylation (OXPHOS) system resides in the mitochondrial cristae membranes and plays a central role in cellular energy transduction. The system is composed of four multimeric respiratory chain enzyme complexes (Complexes I–IV), two electron carriers (ubiquinone and cytochrome-$c$) and a multimeric $F_1F_o$-ATP synthase (also known as Complex V) [1]. The respiratory Complexes I, III and IV form higher-order supercomplexes or respirasomes into flat crista membrane domains, whereas Complex V segregates as dimers into curved crista membrane domains [2]. Collectively, the human OXPHOS complexes are comprised of >90 protein subunits, including 13 subunits encoded in the mitochondrial DNA (mtDNA). In addition, several nuclear-encoded assembly factors are required to build each Complex. Mutations in genes leading to OXPHOS dysfunction cause severe metabolic diseases [3,4], with an estimated prevalence of ~1 in 4,300 [5].

In 1991, Schägger and Von Jagow [6] reported a method for electrophoretic resolution of the five OXPHOS complexes in polyacrylamide gels. The technique uses the mild, nonionic detergent $n$-dodecyl-β-D-maltoside to solubilize membrane proteins without dissociating the individual OXPHOS complexes. The extraction is supported by the addition of the zwitterionic salt 6-aminocaproic acid, which has a zero net charge at pH 7.0 and, hence, does not affect electrophoresis. Coomassie blue G-250 is added to the extracted samples prior to electrophoresis and to the cathode buffer. This anionic blue dye binds to hydrophobic protein surfaces and imposes a negative charge shift on the proteins that forces even basic proteins with hydrophobic domains to migrate towards the anode at pH 7.0. In addition, the induced negative surface charge prevents aggregation of hydrophobic proteins and keeps them soluble in the absence of detergent during electrophoresis. The procedure has been called blue-native polyacrylamide gel electrophoresis (BN-PAGE) because of the characteristic blue Coomassie G-250 dye used during the procedure.

Since the first report, Schägger and others have continued to refine the procedure and have explored new downstream applications. BN-PAGE can be followed by sodium dodecyl sulfate/denaturing polyacrylamide gel electrophoresis (SDS-PAGE) to reveal a two-dimensional pattern of the constituent subunits of the OXPHOS complexes [6–9]. When the very mild, nonionic detergent digitonin, instead of $n$-dodecyl-β-D-maltoside, is used for membrane solubilization, respiratory enzyme supercomplexes remain intact during BN-PAGE, allowing analysis of their composition [10–14]. One-dimensional BN-PAGE and two-dimensional BN/SDS-PAGE have also been combined with conventional western blot analysis and mass spectrometry-based methods to identify component proteins of the complexes [15–18].

Furthermore, it was discovered that established histochemical staining methods can be used to detect in-gel enzymatic activities, since BN-PAGE separates the OXPHOS complexes as intact, catalytically active enzymes [19,20]. More recently, a variant of BN-PAGE, named high-resolution clear-native polyacrylamide gel electrophoresis (CN-PAGE), has been developed, in which the Coomassie blue G-250 dye is replaced by mixtures of anionic and neutral detergents in the cathode buffer [21,22]. Similar to Coomassie blue G-250, the mixed micelles induce a charge shift to membrane proteins to enhance their solubility and augment their electrophoretic migration towards the anode. A key advantage of CN-PAGE is the absence of residual blue dye interference during downstream in-gel enzyme activity staining.

We have used BN-PAGE for >20 years to investigate the assembly of OXPHOS complexes in variety of systems, including cultured fibroblasts [23–27] and skeletal muscle biopsies [28,29] from patients with monogenetic mitochondrial disorders, human cell models [30–32], human peripheral blood mononuclear cells [33], yeast [34] and zebrafish [35]. In addition, we have applied the technique to study the assembly of mitochondrial DNA polymerase γ [36,37] and cytoso-lic ribonucleotide reductase [38]. Although many excellent stepwise protocols of the BN-PAGE technique are available [39–44], we recently posted our step-by-step laboratory protocol online [45]. This protocol is specifically adapted for the analysis of small patient samples and uses a simplified procedure for the extraction of mitochondrial membrane proteins. In contrast to existing protocols, our protocol includes the use of CN-PAGE followed by in-gel enzyme activity staining to avoid interference of residual Coomassie blue G-250 dye and contains an additional enhancement step for in-gel Complex V activity staining.

Here, we validate our protocol with a number of independent experiments. We demonstrate that our method allows the detection of the individual OXPHOS complexes solubilized with *n*-dodecyl-β-ᴅ-maltoside and respiratory supercomplexes solubilized with digitonin. In addition, we show the dynamic range of Complex I, II, IV and V in-gel activity staining. Finally, we use a cell model of Complex I deficiency to test our two-dimensional BN/SDS-PAGE procedure.

## Materials and methods

### Step-by-step protocol for sample preparation, BN- and CN-PAGE, in-gel enzyme activity staining, second dimension SDS-PAGE and western blot analysis

The protocol described in this peer-reviewed article is published on protocols.io, updated May 13, 2025, https://doi.org/10.17504/protocols.io.6qpvrkdrolmk/v1 and is included for printing as supporting information S1 File with this article. Our protocol includes a description of manual casting of native, linear gradient polyacrylamide gels, using the Mini-Protean Tetra Vertical Electrophoresis Cell system (Bio-Rad Laboratories) and a four-way Exponential Gradient Maker (Hoeffer Scientific Instruments, XPO77) connected to a four-way peristaltic pump (Watson Marlow, 205U). Although manual casting offers greater flexibility and is more economical, for greater convenience, precast native 3–12% and 4–16% linear gradient polyacrylamide gels and buffers for BN-PAGE are commercially available from Thermo Fisher Scientific (NativePAGE Bis-Tris gel system). For CN-PAGE, commercial native gels can be combined with the buffers recommended for CN-PAGE in our protocol. Our buffers contain bis-tris as suggested by the first description of the BN-PAGE method [6]. Although bis-tris buffers are compatible with all downstream procedures discussed in the current study, it interferes with commonly used downstream protein determination methods. As an alternative, imidazole-based buffers are recommended [44].

### Cell cultures

A549 human alveolar basal epithelial adenocarcinoma, HEK293T immortalized human embryonic kidney and HeLa S3 human cervical adenocarcinoma cell lines were purchased from the European Collection of Authenticated Cultures. The mtDNA-lacking A549 $\rho^0$ cell line was generated by prolonged cultivation of A549 cells in the presence of 50 ng/ml of ethidium bromide. Absence of mtDNA was verified by PCR. A primary human dermal fibroblast culture was established

from a skin explant of a healthy, 52-y-old female on 9/10/2008 according to standard procedures [46]. Ethical approval was obtained from the Royal Free Hospital and Medical School Research Ethics Committee (reference number: 07/H0720/161). The donor gave prior informed, written consent and all work was carried out in compliance with the Declaration of Helsinki and national legislation.

Cells were cultivated at 37°C in a humidified atmosphere of 5% $CO_2$ in 95% air, in Dulbecco's modified Eagle medium (DMEM) containing GlutaMAX and 25 mM D-(+)-glucose (Gibco Life Technologies, 61965−026), supplemented with 10% fetal bovine serum, 1 mM sodium pyruvate, 0.2 mM uridine (not for HEK293T cells), 50 units/ml of penicillin and 50 µg/ml of streptomycin (culture medium). Cultures were checked regularly for mycoplasma infection. Culture medium of A549, A549 $\rho^0$, fibroblast and HEK293T cultures was refreshed every 3 d and the day before harvesting to prevent nutrient exhaustion; for HeLa S3 cultures, see small interfering RNA (siRNA) transfection procedure below. Confluence was maintained at <90%. Fibroblasts, A549 cells and its $\rho^0$ derivative were expanded in two 10-cm cell culture plates prior to harvesting. This yielded ample material after n-dodecyl-β-D-maltoside extraction to carry out western blot analysis and in-gel activity staining for all OXPHOS complexes. HEK293T cells were expanded in one T75 cell culture flask prior to harvesting. This yielded sufficient material to perform several western blots. To harvest the cultures, cells were dislodged by trypsinization, washed once with culture medium and twice with phosphate-buffered saline (PBS). Cell pellets were obtained by centrifugation, stored at −80°C and used for experiments within 1 week.

### siRNA transfections, SDS-PAGE and western blot analyses of HeLa S3 cells

HeLa S3 cells were transiently transfected with a pair of Qiagen HiPerformance siRNAs targeting the mRNA of nuclear Complex I genes (S1 Table) or scrambled AllStars Negative Control siRNA (Qiagen, 1027281). Prior to transfection, HeLa S3 cells were seeded into wells of a 6-well plate at a density of $3.0 \times 10^5$ cells per well in 2.3 ml of culture medium and returned to the incubator. Transfection mixtures were prepared by mixing 100 µl DMEM with 3.6 µl each of two siRNAs targeting the same gene (10 µM) or 7.2 µl of 10 µM scrambled siRNA and 12 µl of HiPerfect transfection reagent (Qiagen, 391704) in a 1.5-ml tube. After collection of the mixture at the bottom of the tube by a brief centrifugation step, transfection mixtures were incubated at room temperature for 5–10 min and then added dropwise to wells of the 6-well plate with HeLa S3 cells. After 3 d of cultivation, cells were dislodged by trypsinization washed once with culture medium and twice with PBS at 4°C. Cell pellets were stored at −80°C until further analysis.

To reveal the knock down of protein expression by SDS-PAGE and western blot analysis, pellets of transfected and untreated (control) cells were resuspended in 100 µl of pre-cooled 1.5% (w/v) n-dodecyl-β-D-maltoside, PBS, 1 µM PMSF, 1 µg/ml of leupeptin and 1 µg/ml of pepstatin A and incubated on ice for 15 min with regular agitation, followed by centrifugation at 16,000 × g for 15 min at 4°C. Supernatants were collected and protein concentrations were determined with the Pierce BCA Protein Assay Kit (Thermo Fisher Scientific, 23225). Remaining supernatants were stored at −80°C until further analysis. Laemmli Sample Buffer (Bio-Rad Laboratories, 1610747) and NuPAGE Sample Reducing Agent (Invitrogen, NP0009) were added to the samples and after a 10-min incubation at 37°C, samples were resolved by SDS-PAGE, using handcast 15% polyacrylamide gels, followed by western blotting and immunological detection [45]. Precision Plus Protein Standards (Bio-Rad Laboratories, 161-0374) were used as molecular weight markers. Primary and secondary antibodies are listed in S2 and S3 Tables, respectively.

### Preparation of mouse liver mitochondria

The animal study was approved by the UK Home Office (PPL7099586) and carried out in accordance with the University College London Animal Welfare and Ethical Review Body policy. Male mice of the C57BL/6 strain were maintained under standard conditions, with 12-h light-dark cycle, and water and food ad libitum. Husbandry and all procedures followed the guidelines established by the Animal Committee of University College London. The study design and reporting adhered to the Animal Research: Reporting of In Vivo Experiments (ARRIVE) guidelines. To alleviate suffering, any animal exhibiting

persistent marked changes in expected behavior or condition was killed. Mice were sacrificed by exsanguination under general anesthesia with 2% isoflurane in oxygen (Piramal Healthcare). The neck was dislocated to confirm death. Liver was dissected, snap frozen and stored at −80°C. Mouse liver mitochondria were isolated by differential centrifugation as described [41,47] with minor modifications. Briefly, ~5 mg of frozen tissue was excised, washed and minced in saline on ice. The liver tissue was homogenized with a KIMBLE Dounce tissue grinder (Sigma-Aldrich, D8938) in 220 mM mannitol, 70 mM sucrose, 5 mM $KH_2PO_4$, 5 mM $MgCl_2$, 1 mM EGTA, 2 mM HEPES (pH 7.4) on ice. The tissue homogenate was centrifuged at 1,000 x $g$ for 5 min at 4°C to remove cell debris. Supernatants were further centrifuged at 10,000 x $g$ for 10 min at 4°C to pellet mitochondria. Mitochondria were washed twice in the above buffer, followed by storage at −80°C until further analysis.

## Sample preparation for BN- and CN-PAGE

Stored HeLa S3, A549, A549 $\rho^0$ and fibroblast whole cell pellets, and mouse liver mitochondrial pellets were resuspended in 200 μl (100 μl for HeLa S3 cells) of pre-cooled 0.5% (w/v) n-dodecyl-β-D-maltoside, 1 M 6-aminocaproic acid, 50 mM bis-tris and protease inhibitors (pH 7.0), incubated on ice for 15 min with regular agitation, followed by centrifugation at 16,000 × $g$ for 20 min at 4°C and collection of the supernatants (extracts) as described [45]. Protein concentrations of the extracts were determined with the Pierce BCA Protein Assay Kit. Remaining extracts were stored at −80°C until further analysis.

Stored HEK293T whole cell pellets were resuspended in 150 μl of pre-cooled 1 M 6-aminocaproic acid, 50 mM bis-tris and protease inhibitors (pH 7.0), and divided into three equal volumes in separate tubes on ice. To one tube, 5% (w/v) digitonin stock solution (Thermo Fisher Scientific, BN2006) was added to a final concentration of 2%, to one tube, 5% (w/v) digitonin was added to a final concentration of 3%, and to one tube 20% (w/v) n-dodecyl-β-D-maltoside stock solution was added to a final concentration of 0.5%. Tubes were incubated on ice for 15 min with regular agitation, followed by centrifugation at 16,000 × $g$ for 20 min at 4°C and collection of the supernatants (extracts). Protein concentrations of the extracts were determined. Remaining extracts were stored at −80°C until further analysis.

For BN-PAGE of HEK293T, HeLa S3, A549, A549 $\rho^0$, fibroblast and mouse liver mitochondrial extracts, one sixth of a volume of 1 M 6-aminocaprioic acid, 5% (w/v) Coomassie blue G-250 was added to the samples prior to separation in 3–12% or 3–10% (HEK293T extracts) linear gradient gels with a 3% stacking gel [45]. To allow better separation of supercomplexes, BN-PAGE of HEK293T extracts was continued for a further 30 min after the front ran off the gel. For CN-PAGE of A549, A549 $\rho^0$ and mouse liver mitochondrial extracts, one tenth of a volume of 50% (v/v) glycerol and one tenth of a volume of 0.1% (w/v) Ponceau S was added to the samples prior to separation in 3–12% linear gradient gels with a 3% stacking gel [45]. For two-dimensional SDS/BN-PAGE, HeLa cell extracts were first resolved by BN-PAGE, followed by second dimension SDS-PAGE, using 12.5% polyacrylamide gels [45]. Electrophoretic resolution of the samples was followed by western blotting and immunological detection or in-gel enzyme activity staining as described [45].

## BN-PAGE of high molecular weight native protein markers

A vail of the HMW Native Marker Kit (Cytiva, 17044501) was reconstituted in 100 μl of 1 M 6-aminocaproic acid, 50 mM bis-tris (pH 7.0), followed by addition of 17 μl of 1 M 6-aminocaprioic acid, 5% (w/v) Coomassie blue G-250. Two-fold serial dilutions (10 μl/well) were separated by BN-PAGE in a 3–12% linear gradient gel. After electrophoresis, the gel was stained overnight in 0.1% (w/v) Coomassie blue R-250, 40% (v/v) methanol, 10% (v/v) acetic acid, followed by 24-h destaining in water with a paper tissue in the water to absorb the dye.

## Results

### BN-PAGE of native marker proteins

Although BN-PAGE was designed to resolve native membrane proteins, most water-soluble proteins also appear to bind Coomassie blue G-250. As long as the protein is not exceptionally basic, water-soluble proteins will migrate in a discrete

band to the anode during BN-PAGE and fit the log $M_r$-migration distance calibration line [8]. We resolved a set of native, hydrophilic marker proteins by BN-PAGE and confirmed a linear relationship between the log $M_r$ and migration distance (Fig 1), indicating that these marker proteins can be used to estimate the $M_r$ of unknown membrane protein complexes. However, if hydrophilic marker proteins are used to estimate the $M_r$ of membrane protein complexes, the apparent $M_r$ of membrane protein complexes will not correspond to their combined protein moieties, but to their combined protein-lipid compounds.

## Resolution of individual OXPHOS complexes from extracts of cultured cells

To resolve OXPHOS complexes from cultured cells by BN-PAGE, available protocols use either isolated mitochondria [6,15,17,20,22], which requires extensive cell culturing, or treat cells with a low concentration of digitonin to enrich membranous organelles [9,14,39,40,42] before solubilization of membrane proteins with *n*-dodecyl-β-ᴅ-maltoside and addition of Coomassie blue G-250. We found that extracting whole cell pellets in 0.5% (w/v) *n*-dodecyl-β-ᴅ-maltoside, 1 M 6-aminocaproic acid, 50 mM bis-tris to solubilize membrane proteins, followed by centrifugation, yields a supernatant (extract) that can be used directly for BN-PAGE after adding Coomassie blue G-250, or for CN-PAGE after adding glycerol and Ponceau S. This single extraction step is more straightforward and requires less material than procedures that include preliminary isolation of mitochondria or enrichment of membranous organelles.

To test our simplified method, we compared the A549 cell line with its derivative A549 ρ⁰, which lacks mtDNA-encoded subunits of Complex I, III, IV and V, and therefore does not contain these complexes. Cell pellets were extracted with 0.5% *n*-dodecyl-β-ᴅ-maltoside and extracts were subjected to BN- or CN-PAGE, followed by western blotting or in-gel enzyme activity staining to detect the individual complexes. BN-PAGE was used for western blot analysis of Complex I–IV and in-gel Complex I activity staining, whereas CN-PAGE was used for western blot analysis of Complex V, and in-gel Complex II and V activity staining.

In the A549 sample, Complex I (970 kDa) was detected as a single band in the upper part of a western blot probed with an antibody against the nuclear-encoded Complex I subunit NDUFB6 (Fig 2). This band was absent in the A549 ρ⁰ sample, but we did detect a band at the bottom of the blot, which we assume represents free, unassembled, 15-kDa NDUFB6. To detect Complex I activity, the cathode buffer containing Coomassie blue G-250 dye was substituted for colorless cathode buffer without dye after 15 min BN-PAGE [45], to reduce interference of the blue dye during subsequent enzyme activity staining. In our experience, this procedure results in better resolution of Complex I than CN-PAGE. In-gel NADH

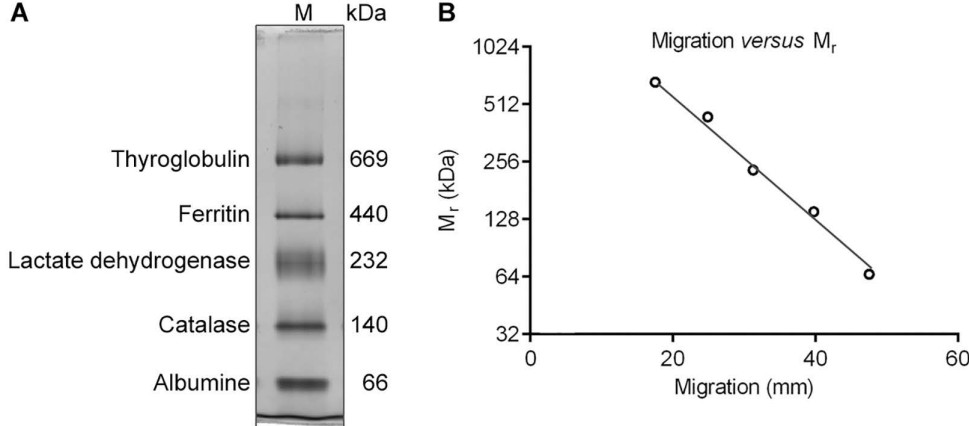

**Fig 1. BN-PAGE of native marker proteins. (A)** Coomassie Brilliant blue R-250-stained gel of native, water-soluble marker proteins separated by BN-PAGE. **(B)** The log $M_r$-migration distance calibration line.

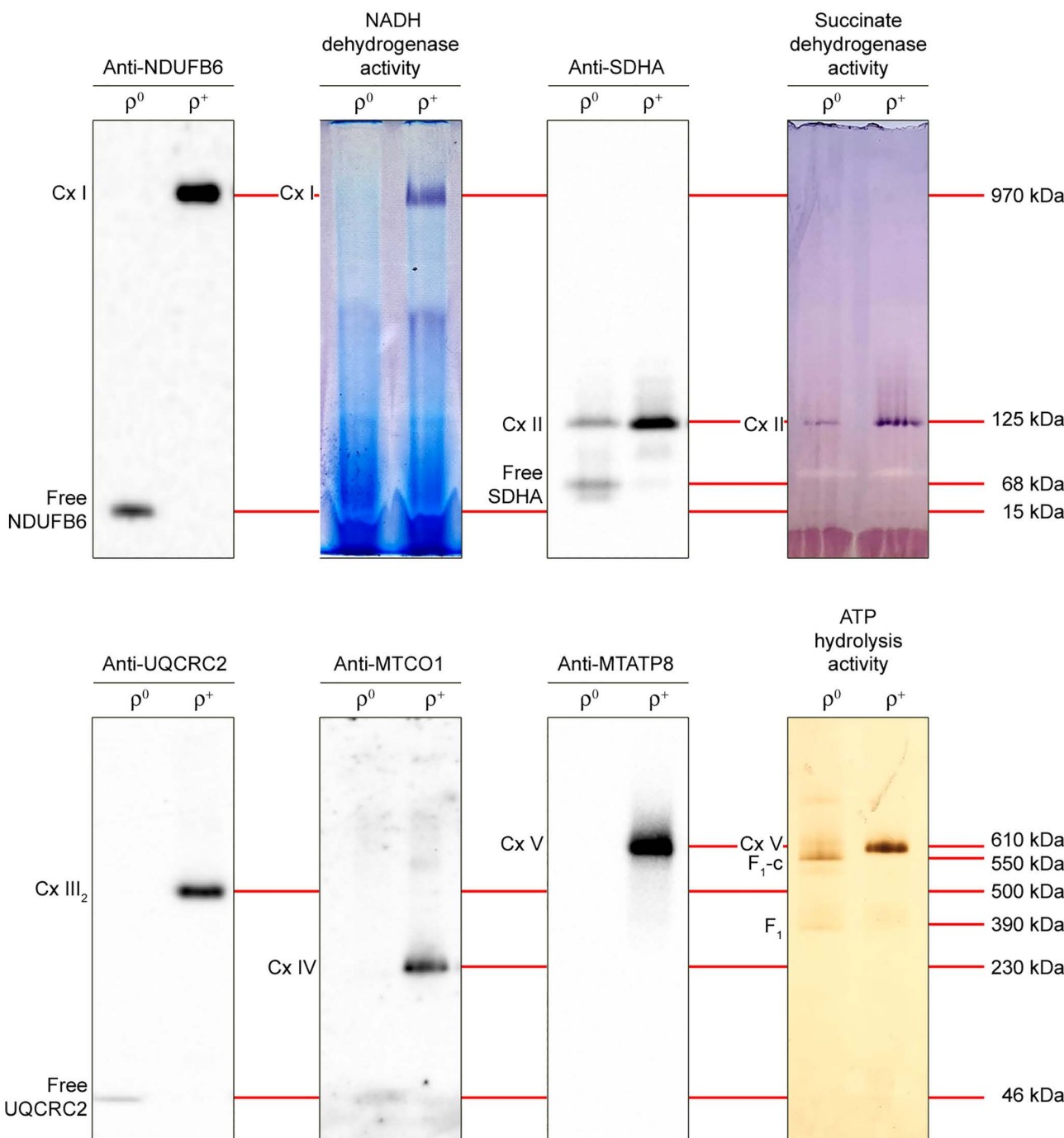

**Fig 2. Western blot and in-gel activity detection of individual OXPHOS complexes from A549 cells.** A549 ($\rho^+$) and A549 $\rho^0$ ($\rho^0$) *n*-dodecyl-β-ᴅ-maltoside extracts were resolved by 3–12% BN- or CN-PAGE, followed by western blot analysis with indicated antibodies or in-gel staining for indicated enzyme activities. BN-PAGE was used to detect Complex I (Cx I), Complex II (Cx II), Complex III dimer (Cx III$_2$) and Complex IV (Cx IV) on western blots (10 μg of protein extract per lane) and for in-gel Complex I activity (30 μg of protein extract per lane). CN-PAGE was used to detect Complex V (Cx V) on a western blot (5 μg of protein extract per lane) and for in-gel Complex II and V activity (30 μg and 10 μg of protein extract per lane, respectively). F$_1$-c and F$_1$ denote the F$_1$-portion of Complex V associated with the ring of c subunits and the F$_1$-portion on its own, respectively. Approximate molecular weights are indicated.

dehydrogenase activity staining revealed a violet band in the A549 sample that co-migrated with Complex I detected with anti-NDUFB6 antibody and represents Complex I activity. As expected, this violet band was absent in the A549 $\rho^0$ sample.

Complex II (125 kDa) was detected near the bottom of a western blot probed with an antibody against Complex II subunit SDHA (Fig 2). The Complex II band was present in the A549 as well as the A549 $\rho^0$ sample, but the signal was much fainter for the A549 $\rho^0$ sample, suggesting a partial Complex II deficiency in the mtDNA-lacking A549 $\rho^0$ cells, even though all four Complex II subunits are nuclear-encoded. In addition to the Complex II band, a faster migrating band was detected at the bottom of the blot in the A549 $\rho^0$ sample. We assume that this band represents unassembled, 68-kDa SDHA, suggesting Complex II assembly is disturbed in A549 $\rho^0$ cells. In-gel succinate dehydrogenase activity staining revealed a violet band that co-migrated with Complex II detected with anti-SDHA antibody and represents Complex II activity. Complex II activity was lower in the A549 $\rho^0$ sample than in the A549 sample, matching the Complex II immunological detection.

Complex III functions as a tightly linked, 500-kDa dimer [48,49]. The dimer was detected in the A549 sample in the middle of a western blot probed with an antibody against the nuclear-encoded Complex III subunit UQCRC2, migrating between 230-kDa Complex IV and 610-kDa Complex V (Fig 2). Similar to Complex I, the Complex III dimer (III$_2$) was not detected in the A549 $\rho^0$ sample, but this sample did show a faint band at the bottom of the blot, which we assume represents unassembled, 46-kDa UQCRC2. Although in-gel Complex III activity staining has been reported [21], we have been unable to stain gels for Complex III activity.

Complex IV was detected in the A549 sample just below the middle of a western blot probed with an antibody against the mtDNA-encoded Complex IV subunit MTCO1 (Fig 2). As expected, this band was not present in the A549 $\rho^0$ sample. So far, we and others [22,42] have been unable to stain gels with cultured cell fractions for Complex IV activity in a conclusive manner.

Complex V consists of two functional domains, $F_1$ and $F_o$. The $F_1$-domain protrudes into the mitochondrial matrix and contains the ATP synthase/hydrolase activity, whereas the $F_o$-domain is embedded in the cristae membrane and contains the proton transfer activity [50]. For the resolution of holo-Complex V, we prefer CN-PAGE because we found that Coomassie blue G-250 dye results in partial dissociation of the $F_1$- and $F_o$-part in some samples (S1 Fig), as has been noted by others [22,51]. After CN-PAGE, Complex V was detected just above the middle of a western blot probed with an antibody against the mtDNA-encoded Complex V $F_o$-domain subunit MTATP8 in the A549 sample but not in the A549 $\rho^0$ sample (Fig 2). For in-gel Complex V activity staining we used a brief enhancement step, in which by the addition of ammonium sulfide solution, the white lead(II) phosphate precipitate is converted into a dark brown lead(II) sulfide precipitate [33]. To our knowledge, this extra step has only been used by others for in-gel activity staining of chloroplast $F_1F_o$-ATP synthase [52]. In-gel ATP hydrolysis activity staining revealed a dark brown band in the A549 sample that co-migrated with Complex V detected with anti-MTATP8 antibody and represents Complex V activity. In the A549 $\rho^0$ sample, we detected a faint band migrating slightly faster than 610-kDa Complex V. We assume that this band represents the ~550-kDa band identified by Wittig et al. [53] as the $F_1$-portion associated with at least the ring of c (ATP5G1–3) subunits but probably most of the other subunits comprising the $F_o$-portion, except for the mtDNA-encoded subunits MTATP6 and MTATP8. In addition, we detect a faster migrating, very faint band in A549 $\rho^0$ cells that we assume represents the single 390-kDa $F_1$-portion.

### Resolution of individual OXPHOS complexes from mouse liver mitochondria

We repeated the experiments with an *n*-dodecyl-β-ᴅ-maltoside extract of mouse liver mitochondria (Fig 3). BN-PAGE was used for western blot analysis of Complex I–IV and in-gel Complex I activity staining, while CN-PAGE was used for western blot analysis of Complex V, and Complex II, IV and V in-gel activity staining. Complex I was detected in the upper part of a western blot probed with anti-NDUFB6 antibody and co-migrated with in-gel Complex I activity. Complex II was detected near the bottom of a western blot probed with anti-SDHA antibody and co-migrated with in-gel Complex II activity. In addition, free, unassembled SDHA was detected as a prominent band at the bottom of the blot, suggesting an

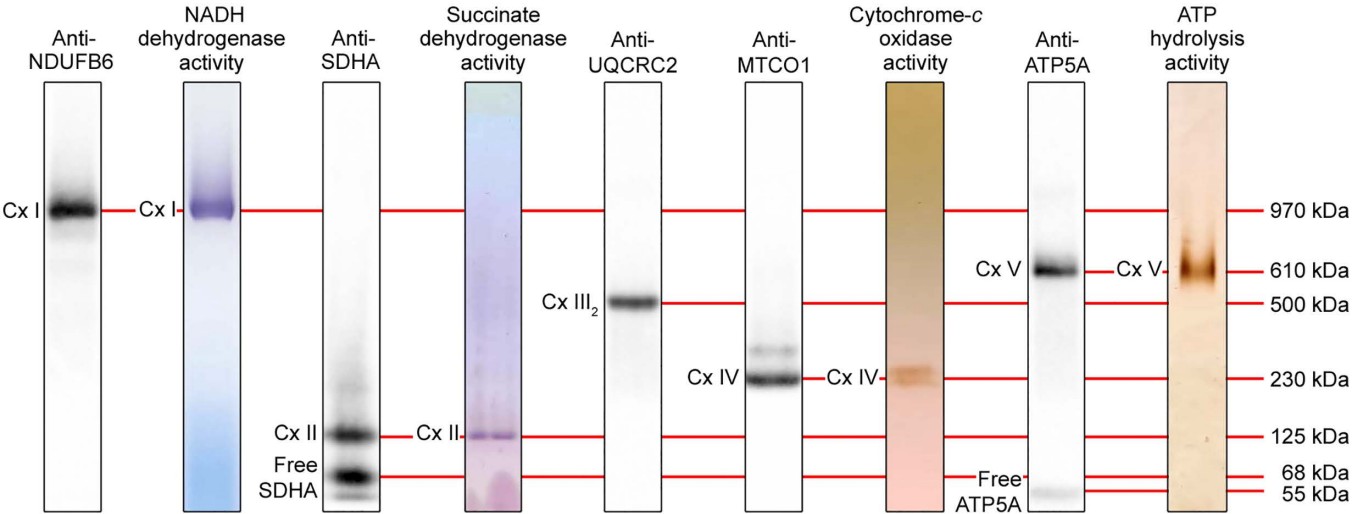

**Fig 3. Western blot and in-gel enzyme activity detection of individual OXPHOS complexes from mouse liver mitochondria.** Mouse liver mitochondrial *n*-dodecyl-β-ᴅ-maltoside extracts were resolved by 3–12% BN- or CN-PAGE (10 µg/lane, but 20 µg/lane for in-gel Complex IV activity staining), followed by western blot analysis with indicated antibodies or in-gel activity staining for indicated enzymes. BN-PAGE was used to detect Complex I (Cx I), Complex II (Cx II), Complex III dimer (Cx III$_2$) and Complex IV (Cx IV) on western blots and for in-gel Complex I activity. CN-PAGE was used to detect Complex V (Cx V) on a western blot and for in-gel Complex II, IV and V activity. Approximate molecular weights are indicated.

excess of this subunit in mouse liver mitochondria. Complex III$_2$ was detected in the middle of the blot probed with anti-UQCRC2 antibody, migrating between Complex IV and Complex V. Complex IV was detected just below the middle of the blot probed with anti-MTCO1 antibody and co-migrated with in-gel Complex IV activity. In contrast to cultured cell extracts, mitochondrial extracts show a clear in-gel Complex IV activity after prolonged incubation with reaction buffer. Finally, Complex V was detected just above the middle of a western blot probed with an antibody against the nuclear-encoded Complex V F$_1$-domain subunit ATP5A and co-migrated with in-gel Complex V activity. In addition, unassembled ATP5A was detected as a faint band at the bottom of the blot, however, the F$_1$-subassembly was not detected.

## Determination of the dynamic range of in-gel enzyme activity staining

To determine the dynamic range for in-gel enzyme activity staining, we resolved serial dilutions of *n*-dodecyl-β-ᴅ-maltoside extracted A549 cells by BN- or CN-PAGE, followed by in-gel enzyme activity staining for Complex I, II and V (Fig 4A). A serial dilution of *n*-dodecyl-β-ᴅ-maltoside extracted mouse liver mitochondria was used for in-gel Complex IV activity staining (Fig 4B) because this staining is not sensitive enough to detect Complex IV activity of cultured cell extracts. The staining revealed a practically linear relationship between loading and staining intensity over a wide range for in-gel Complex I, II and V activity staining (Fig 4C, 4D, 4F). Detection limits were 5 µg of A549 protein extract per lane for Complex I, 10 µg of A549 protein extract per lane for Complex II and 2.5 µg of A549 protein extract per lane for Complex V. In contrast, the dynamic range of in-gel Complex IV activity staining was much more limited than in-gel Complex I, II and V activity staining (Fig 4E). In-gel Complex IV activity staining had a detection limit of 5 µg of mouse liver mitochondrial extract per lane.

## Resolution of respiratory chain enzyme supercomplexes

To detect Complex I, III and IV higher-order supercomplexes, HEK293T cell pellets were extracted with digitonin, which has weaker delipidating properties and is less likely to dissociate labile hydrophobic interactions than *n*-dodecyl-β-ᴅ-maltoside. For comparison, HEK293T samples extracted with 0.5% *n*-dodecyl-β-ᴅ-maltoside were loaded alongside 2%

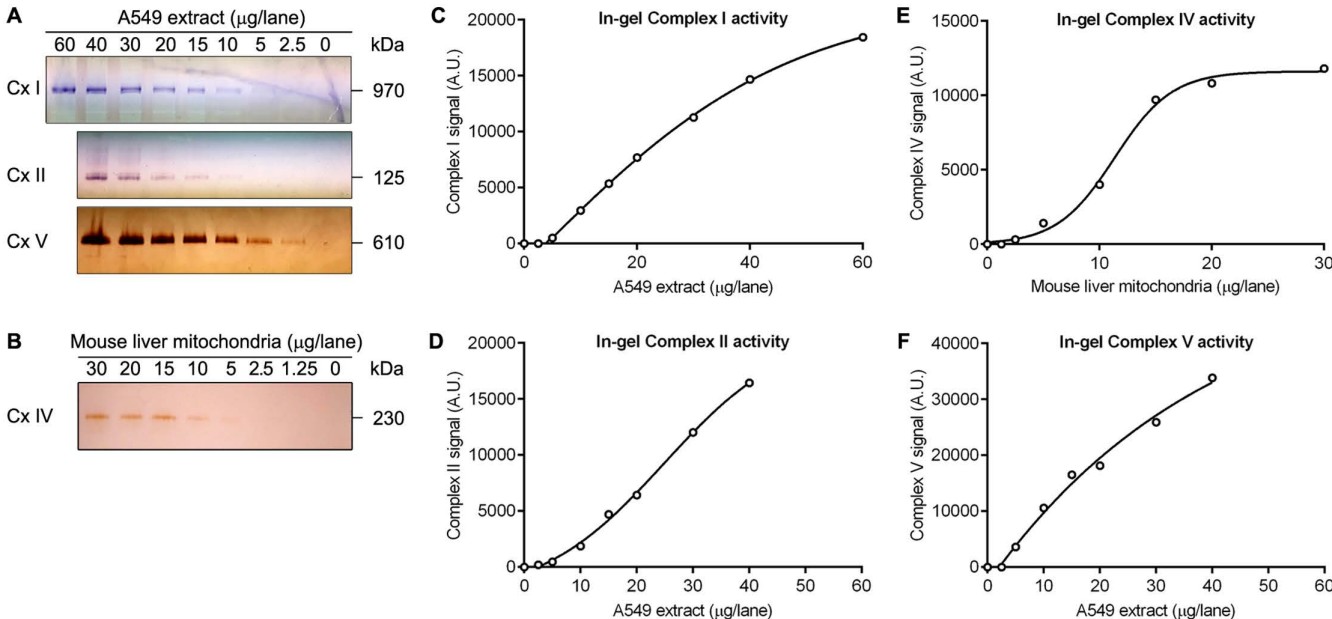

**Fig 4. In-gel enzyme activity staining of serial dilutions of extracts from A549 cells and mouse liver mitochondria.** *N*-dodecyl-β-ᴅ-maltoside extracts were resolved by 3–12% BN- or CN-PAGE, followed by in-gel activity staining for indicated enzymes. BN-PAGE was used to resolve Complex I (Cx I), CN-PAGE was used to resolve Complex II (Cx II), Complex IV (Cx IV) and Complex V (Cx V). **(A)** In-gel activity staining for Complex I, II and V in A549 whole-cell extracts. **(B)** In-gel activity staining for Complex IV in mouse liver mitochondrial extracts. **(C)** Relationship between loading and in-gel Complex I activity signal. **(D)** Relationship between loading and in-gel Complex II activity signal. **(E)** Relationship between loading and in-gel Complex IV activity signal. **(F)** Relationship between loading and in-gel Complex V activity signal.

and 3% digitonin extracts in triplicate on a 3–10% native gel and subjected to BN-PAGE followed by western blotting. The blot was cut to create three identical blots, each with the three samples. Blots were probed with anti-NDUFB6 (Complex I), anti-UQCRC2 (Complex III) or anti-MTCO1 (Complex IV) antibodies. The three blots were aligned to visualize the co-migration of the complexes, indicating the constituents of the higher-order complexes. Although there was a clear difference between the pattern of immunoreactive bands in the *n*-dodecyl-β-ᴅ-maltoside extract and the digitonin extracts, there were no overt differences between the two digitonin extracts (Fig 5). However, in a preliminary experiment, extraction with 1% digitonin resulted in western blots with high molecular weight smears of immunoreactive material (S2 Fig), indicating that 1% digitonin is insufficient to solubilize the respiratory chain complexes.

The anti-NDUFB6 antibody detected singular Complex I in the *n*-dodecyl-β-ᴅ-maltoside extract, and two slower migrating bands near the top of the blot in both digitonin extracts (Fig 5). The anti-UQCRC2 antibody detected Complex III$_2$ in the *n*-dodecyl-β-ᴅ-maltoside extract, and a slightly slower migrating band in the digitonin extracts. We assume that the latter band also represents Complex III$_2$, migrating slightly slower due to detergent-dependent increased amounts of boundary lipids. In addition, the anti-UQCRC2 antibody recognized two bands near the top of the blot in the digitonin extracts that comigrated with bands recognized by the anti-NDUFB6 antibody. We presume that these two bands represent higher-order complexes of at least Complex I and Complex III$_2$.

The anti-MTCO1 antibody revealed singular Complex IV in the *n*-dodecyl-β-ᴅ-maltoside extract as a very prominent band in the lower part of the blot and a faint band just above the middle of the blot (Fig 5). Furthermore, the anti-MTCO1 antibody recognized four bands in the digitonin extracts. We believe that the most prominent, fastest migrating band in the digitonin extracts also represents singular Complex IV. Singular Complex IV in the digitonin extract is expected to migrate slower than singular Complex IV in the *n*-dodecyl-β-ᴅ-maltoside extract because the

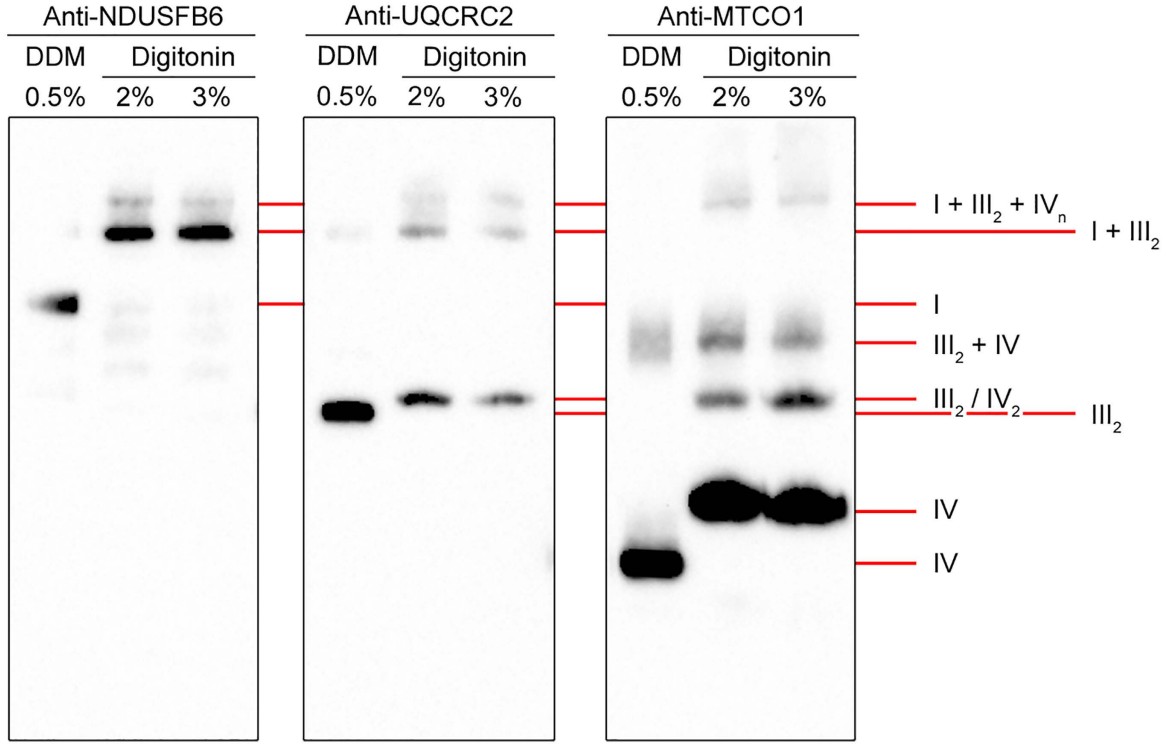

**Fig 5. Western blot detection of respiratory chain enzyme supercomplexes from HEK293T cells.** HEK293T cell suspensions were extracted with *n*-dodecyl-β-d-maltoside (DDM) or digitonin at specified detergent concentrations, followed by 3–10% BN-PAGE (10 μg protein extract/lane) and western blot analysis with indicated antibodies. Migrations of individual respiratory chain enzyme complexes and higher-order complexes are indicated.

digitonin extract will contain the 14-subunit holo-Complex IV [18,54], whereas the *n*-dodecyl-β-D-maltoside extract will contain a 13-subunit Complex IV, lacking the 9-kDa subunit COXFA4, as this subunit dissociates in the presence of 0.5% *n*-dodecyl-β-D-maltoside [28,55]. Moreover, digitonin-extracted singular Complex IV is expected to contain more boundary lipids than the *n*-dodecyl-β-D-maltoside-extracted enzyme. We think that the second fastest migrating band recognized with the anti-MTCO1 antibody in the digitonin extracts represents a Complex IV dimer, which is known to co-migrate with Complex III$_2$ [40,41]. The third fastest migrating band recognized with the anti-MTCO1 antibody in the digitonin extracts most likely represents a supercomplex of Complex IV and Complex III$_2$ [40,41]. Unexpectedly, we do not detect this supercomplex with the anti-UQCRC2 antibody, probably because its level is below the detection limit. The faint band in the *n*-dodecyl-β-D-maltoside extract recognized by the anti-MTCO1 antibody migrates marginally faster than the Complex III$_2$–IV supercomplex. Possibly, this faint band also represents the Complex III$_2$–IV supercomplex but lacks COXFA4 and holds fewer boundary lipids, therefore migrating slightly faster. The slowest migrating band recognized by the anti-MTCO1 antibody in the digitonin extracts comigrates with the slowest migrating bands recognized by the anti-NDUFB6 and anti-UQCRC2 antibodies, indicating that this band represents the respirasome consisting of Complex I, III$_2$ and IV [56,57].

Remarkably, we do not detect singular Complex I in the digitonin extracts; all detectable Complex I appears part of higher-order structures with Complex IV and/or Complex III$_2$ (Fig 5). This is compatible with earlier publications, which reported that some tissues, including HEK293T cells, do not show any detectable singular Complex I when extracted with digitonin [40,58,59].

## Western blot analysis of Complex I subassemblies after one-dimensional BN-PAGE and two-dimensional BN/SDS-PAGE

Complex I is a 970-kDa enzyme composed of 44 different protein subunits [18,60]. The enzyme has an L-shaped structure with one arm projecting into the mitochondrial matrix and one arm inserted into the cristae membrane [61,62]. The matrix arm comprises the N- and Q-module, where the substrate NADH is oxidized and its electrons are transferred to ubiquinone, respectively. The membrane arm encompasses the P-module, where protons are pumped from the matrix to the intermembrane space. The assembly process of Complex I involves the stepwise merging of discrete subcomplexes, each composed of multiple subunits, that converge in a temporally and spatially coordinated fashion [63,64]. To validate our two-dimensional BN/SDS-PAGE method, we studied a late step of the modular assembly process by knocking down the expression of subunits NDUFV1, NDUFV2 and NDUFS4 of the N-module and subunit NDUFS2 of the Q-module through transient siRNA transfection of HeLa S3 cell cultures.

HeLa S3 cell cultures were transfected with pairs of siRNA species targeting *NDUFV1*, *NDUFV2*, *NDUFS2* or *NDUFS4* gene expression to ensure efficient knock down. Each gene was targeted with two different pairs of siRNAs in separate transfection experiments to confirm the reproducibility of the results. Untransfected cells and cells transfected with scrambled siRNA were used as controls. Cells were harvested 3 d after transfection, followed by SDS-PAGE and western blotting to determine the expression levels of NDUFV1, NDUFV2, NDUFS2 or NDUFS4. Blots were re-probed with an antibody against Complex II subunit SDHA to verify even protein loading of the samples. The analyses showed that each siRNA transfection resulted in a clear knock down of the expression of the targeted Complex I subunit (Fig 6).

To reveal possible Complex I subassemblies in the knocked down cells, the siRNA transfections were repeated. Cells were harvested 3 d post-transfection, followed by BN-PAGE and western blot analysis. Probing of a western blot with an antibody against NDUFV1 demonstrated lower levels of Complex I in all siRNA-transfected cultures compared to the untransfected control culture (Fig 7A). In addition, we detected a band, labeled "*b*" with anti-NDUFV1 antibody that comigrated with Complex III$_2$. We assume that this is a non-specific band because the signal was comparable in all samples. We also detected a band labeled "*c*" with anti-NDUFV1 antibody that migrated slightly faster than Complex II. This band was very faint in untransfected control and NDUFS2 knock down cells but was very strong in NDUFS4 knock down cells. Probing of a western blot with an anti-NDUFV2 antibody produced a very similar result as the probing with anti-NDUFV1, except that band *b* was not detected (Fig 7A). Probing of blots with anti-NDUFS2, anti-MTND1, anti-NDUFA9 or anti-NDUFB6 antibodies gave comparable results in all four cases (Fig 7A). Generally, the

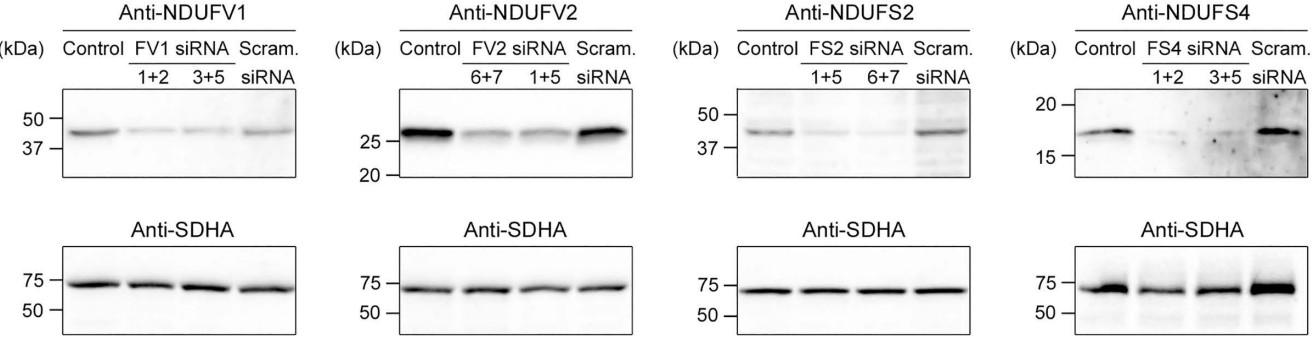

**Fig 6. Knock down of Complex I gene expression by siRNA transfection.** HeLa S3 cells were transfected with pairs of siRNA species targeting *NDUFV1* (FV1), *NDUFV2* (FV2), *NDUFS2* (FS2) or *NDUFS4* (FS4) mRNA, or scrambled (scram.) siRNA. Untransfected HeLa S3 cells were included as a control. After 3 d, samples were resolved by SDS-PAGE (10 μg protein/lane), followed by western blot analysis with indicated antibodies. Blots were re-probed with anti-SDHA antibody to confirm equal loading. Migration of molecular weight markers is indicated.

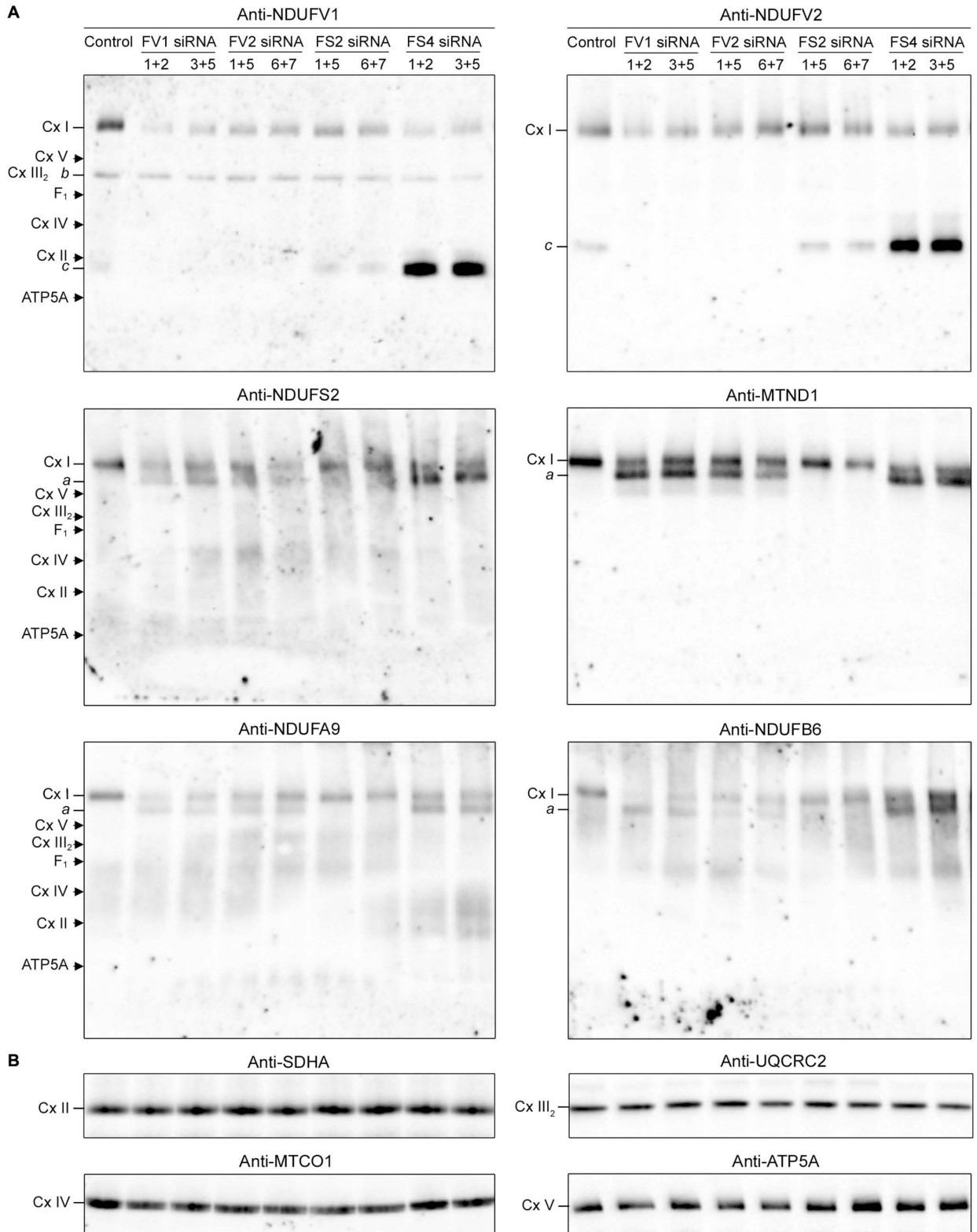

**Fig 7. Complex I subassemblies after knocking down of Complex I gene expression.** HeLa S3 cells were transfected with two pairs of siRNA species targeting *NDUFV1* (FV1), *NDUFV2* (FV2), *NDUFS2* (FS2) or *NDUFS4* (FS4) mRNA. Untransfected HeLa S3 cells were included as control.

After 3 d, samples were resolved by BN-PAGE (10 μg protein/lane), followed by western blot analysis with indicated antibodies. Blots were re-probed with antibodies against Complexes II–V. **(A)** Western blots probed with antibodies against Complex I subunits. Migrations of Complex I–V, the $F_1$-moiety of Complex V ($F_1$) and unassembled ATP5A are indicated. Newly identified bands are labeled *a*, *b* and *c*. **(B)** Western blots re-probed with antibodies against subunits of Complex II–V.

siRNA-transfected cultures showed lower levels of Complex I than the untransfected control culture. In addition, we detected a band labeled "*a*" with these four antibodies that migrated slightly faster than Complex I in cultures in which NDUFV1, NDUFV2 or NDUFS4 expression was knocked down. Blots were re-probed with anti-SDHA, anti-UQCRC2, anti-MTCO1 and anti-ATP5A antibodies to detect Complex II, III, IV and V, respectively. The re-probing demonstrated that Complexes II–V were not affected by the knock down of Complex I subunits ([Fig 7B]). In addition, the re-probing provided convenient molecular weight markers.

Western blot analysis of the knocked down HeLa S3 cultures resolved by BN-PAGE revealed three 'new' bands: *a*, *b* and *c*. Band *a*, which is detected in NDUFV1, NDUFV2 and NDUFS4 knock down cultures and migrates a bit slower than 970-kDa Complex I, most likely represents the 770-kDa subassembly consisting of the Q- and P-module, because: (1) band *a* is detected with antibodies against constituent subunits of the Q- or P-module (anti-NDUFS2, anti-NDUFA9, anti-MTND1 and anti-NDUFB6) but not with antibodies against constituent subunits of the N-module (anti-NDUFV1 and anti-NDUFV2), and (2) band *a* is detected when constituent subunits of the N-module are knocked down (NDUFV1, NDUFV2 and NDUFS4) but not when a constituent subunit of the Q-module is knocked down (NDUFS2). As mentioned above, we think that band *b* represents a nonspecific signal. Band *c*, however, most likely represents the partly pre-assembled N-module of 160 kDa, consisting of NDUFV1, NDUFV2, NDUFS1 and NDUFA2, identified by Guerrero-Castillo and colleagues [63], because (1) band *c* is detected with antibodies against constituent subunits of this partly pre-assembled N-module (anti-NDUFV1 and anti-NDUFV2) but not with antibodies against constituent subunits of the Q- or P-module (anti-NDUFS2, anti-NDUFA9, anti-MTND1 and anti-NDUFB6), (2) band *c* is detected when NDUFS4 is knocked down; NDUFS4 is a constituent subunit of the N-module but not part of the 160-kDa partly pre-assembled N-module, and (3) band *c* is detected when NDUFS2 is knocked down; NDUFS2 is a constituent subunit of the Q-module. Knock down of NDUFS2 is likely to prevent assembly of the Q- and P-module and, consequently, stalls further assembly of the partly pre-assembled N-module. The high intensity of band *c* in NDUFS4 knock down cultures compared to the low intensity of band *c* in NDUFS2 knock down cultures suggests that the partly pre-assembled 160-kDa N-module is overexpressed in NDUFS4 knock down cultures, possibly as a compensatory response to NDUFS4 loss.

To confirm the component subunits of the Complex I subassemblies *a* and *c*, we performed two-dimensional BN/SDS-PAGE followed by western blot analysis of an untransfected control sample and an NDUFS4 knock down sample. Western blots were probed with a cocktail of anti-NDUFV1, anti-NDUFV2, anti-MTND1 and anti-NDUFB6 antibodies. Subunit spots on the two-dimensional blot were identified according to their molecular weight. The two-dimensional blot with the untransfected control sample confirmed that NDUFV1, NDUFV2, MTND1 and NDUFB6 were present in holo-Complex I detected on one-dimensional BN-PAGE blots ([Fig 8A]). The two-dimensional blot with the NDUFS4 knock down sample showed that band *a* contained P-module subunits MTND1 and NDUFB6 but not N-module subunits NDUFV1 and NDUFV2 ([Fig 8A]), which is consistent with the view that band *a* represents the large Q- and P-module subassembly ([Fig 8B]). In addition, this blot indicated that band *c* contains N-module subunits NDUFV1 and NDUFV2 but not P-module subunits MTND1 and NDUFB6 ([Fig 8A]), which is consistent with the view that band *c* represents the partly pre-assembled N-module. Both two-dimensional blots show four additional spots labeled "α", "β", "γ" and "δ" in [Fig 8A]. We assume that these are nonspecific signals; spots α and β do not align with bands *a* and *b* on the one-dimensional blots, and the apparent molecular weights of spots γ and δ do not correspond with any of the probed subunits.

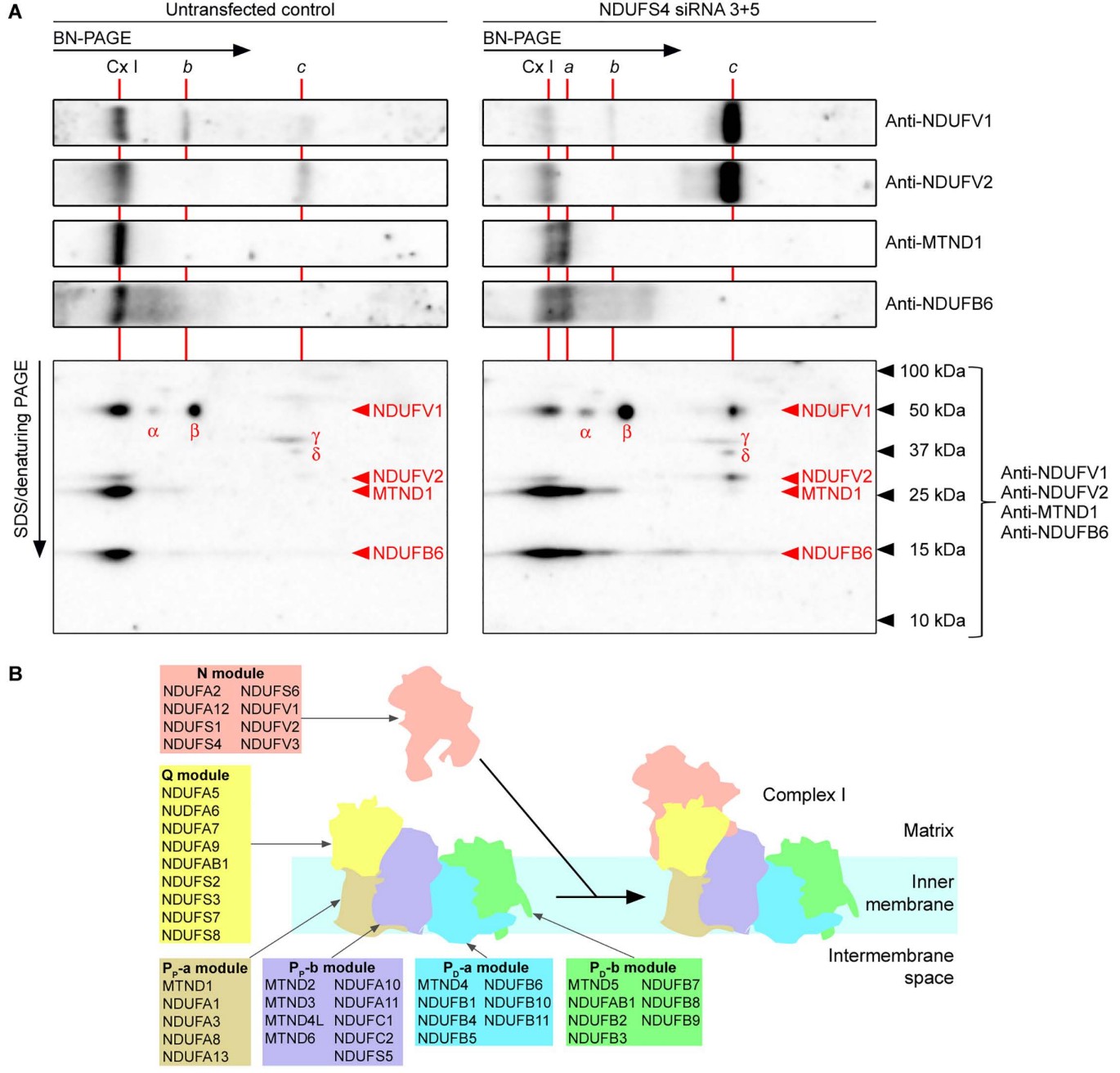

**Fig 8. Identification of component subunits of Complex I subassemblies.** HeLa S3 cells were transfected with a pair of siRNA species targeting *NDUFS4* mRNA. Untransfected HeLa S3 cells were included as control. After 3 d, samples were resolved by BN-PAGE (10 μg protein/lane), followed by second dimension SDS-PAGE and western blot analysis with indicated antibodies. **(A)** One-dimensional BN-PAGE western blot strips and two-dimensional SDS/BN-PAGE western blots probed with antibodies against Complex I subunits. Complex I subunits were identified according to their molecular weight. SDS-PAGE migrations of marker proteins are indicated. Newly identified bands on the one-dimensional blots are labeled *a*, *b* and *c*. Likely non-specific spots on the two-dimensional blots are labeled α, β, γ and δ. **(B)** Subunit composition of the Complex I modules and final step of Complex I assembly.

## Discussion

Recently, we published our stepwise BN- and CN-PAGE laboratory protocol for the characterization of OXPHOS enzyme complexes [45]. The protocol is tailored for the analysis of small tissue or cell culture samples and describes the use of different cathode buffers in combination with BN- or CN-PAGE to achieve optimal downstream in-gel enzyme activity staining or western blot results (Fig 9). The method includes a streamlined sample preparation method, which involves extraction of whole cell pellets with $n$-dodecyl-β-D-maltoside. This procedure saves time and requires less starting material compared to published methods, which use isolated mitochondria or treat the cells with digitonin to enrich membranous organelles prior to solubilization of membrane proteins with $n$-dodecyl-β-D-maltoside. In addition, we apply an enhancement step during the in-gel Complex V staining, which improves the sensitivity of the assay dramatically.

In the current study, we validated our procedure. First, we showed that, although originally developed for the separation of hydrophobic membrane proteins, BN-PAGE can also be used for the separation of hydrophilic, water-soluble proteins to provide a linear log $M_r$ versus migration distance calibration line. To verify the specificity of the detection of individual OXPHOS complexes on western blots or by in-gel enzyme activity staining, we compared A549 cell samples with those from its mtDNA-deficient derivative A549 $\rho^0$. A549 $\rho^0$ cells do not contain fully assembled Complex I, III, IV and V, as some of the subunits of these complexes are encoded by mtDNA. Comparison of A549 samples with the negative control A549 $\rho^0$ samples confirmed that our method detects assembled OXPHOS complexes.

In-gel enzyme activity staining provides semi-quantitative results. Nevertheless, we found that in-gel Complex I, II and V activity staining show a wide dynamic range. In our experience, in-gel activity staining for these Complexes is more sensitive than spectrophotometric activity assays. This is particularly true for in-gel Complex V activity staining after the enhancement step with ammonium sulfide solution, which increases sensitivity considerably. In contrast to in-gel Complex I, II and V

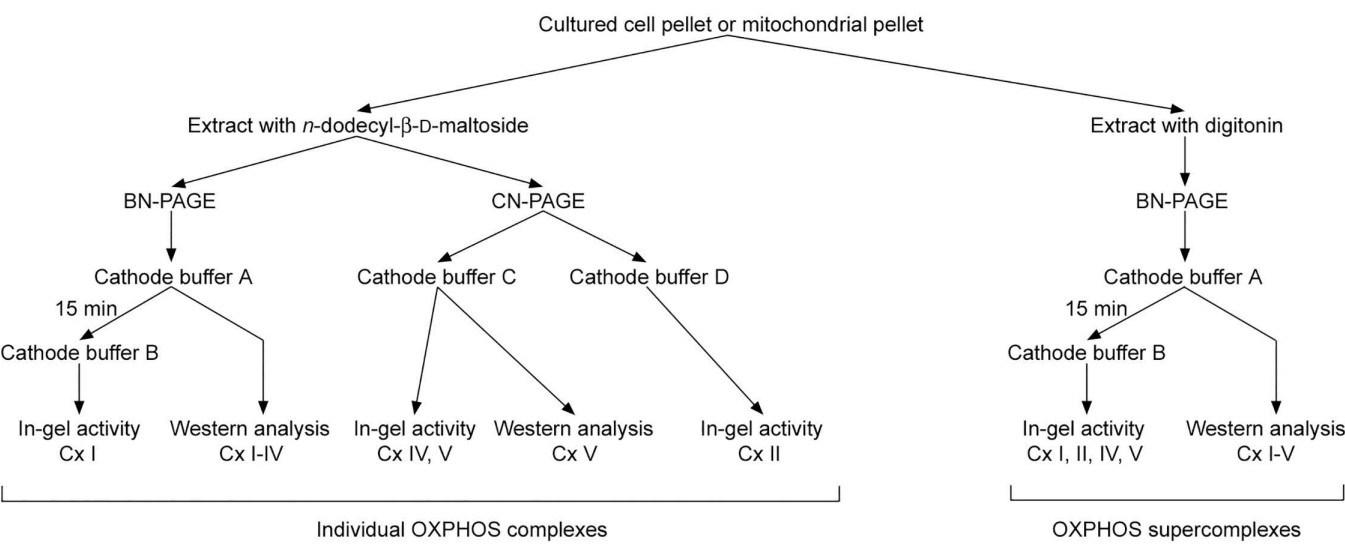

Cathode buffer A: 50 mM tricine, 15 mM bis-tris, 0.02% (w/v) Coomassie blue G-250 (pH7.0)
Cathode buffer B: 50 mM tricine, 15 mM bis-tris (pH7.0)
Cathode buffer C: 50 mM tricine, 15 mM bis-tris, 0.05% (v/v) Triton X-100, 0.05% (w/v) sodium deoxycholate (pH7.0)
Cathode buffer D: 50 mM tricine, 15 mM bis-tris, 0.02% (w/v) $n$-dodecyl-β-D-maltoside, 0.05% (w/v) sodium deoxycholate (pH7.0)

**Fig 9. Schematic presentation of the workflow.** Extraction and resolution of individual or supercomplexes of Complex I–V (Cx I–V) from pellets of cultured cells or isolated mitochondria by BN- or CN-PAGE, followed by in-gel enzyme activity staining or western blot analysis. Note that different cathode buffers are recommended for specific applications.

activity staining, in-gel Complex IV activity staining is not sufficiently sensitive to be used for whole-cell extracts, though the staining can be used to reveal Complex IV activity of mouse liver mitochondrial extracts. Given the liver's well-established high mitochondrial density, and elevated metabolic activity and OXPHOS enzyme content compared to cultured cells, these inherent biological characteristics are likely to contribute to the differential staining outcomes. In addition, in-gel Complex IV activity staining shows a narrow dynamic range. Therefore, in parallel to in-gel activity staining, we recommend carrying out spectrophotometric or polarographic Complex IV assays, which are more sensitive and produce quantitative data.

Typically, we load gels with 10 µg of sample for western blot analysis. If the sample amount is limited or the antibodies are very sensitive, as, e.g., is the case with anti-ATP5A used to detect Complex V, then half the amount of sample can be loaded. In-gel Complex I and II activity staining requires 30 µg of whole cell extract per lane but 10 µg is sufficient for mito-chondrial samples because these samples are more enriched for OXPHOS complexes. As the in-gel Complex IV activity staining is quite insensitive, we recommend using 20 µg of mitochondrial sample per lane and a prolonged incubation with cytochrome-*c* solution. On the other hand, in-gel Complex V staining is very sensitive and requires only 5 µg of sample per lane.

The detergent to protein ratio is critical for solubilizing membrane proteins. The recommended ratio of *n*-dodecyl-β-D-maltoside to protein ranges from 1 to 2.5 g/g [40,44]. In titration experiments, we showed that an excess of *n*-dodecyl-β-D-maltoside has no apparent effect on the integrity of the individual OXPHOS complexes, except that subunit COXFA4 dissociates from Complex IV [28]. In our experience, solubilization of a cell pellet derived from two 10-cm culture plates in 200 µl extraction buffer containing 0.5% *n*-dodecyl-β-D-maltoside results in a near complete extraction of the mitochondrial content and solubilization of the individual OXHOS complexes. Although an excess of *n*-dodecyl-β-D-maltoside does not seem to have a detrimental effect, we recommend performing a titration to optimize the procedure prior to analysis of a new cell line or tissue. Schägger and Pfeiffer [11] originally suggested a ratio of digitonin to protein of 4 g/g and later recommended 6 g/g [44] to resolve respiratory chain enzyme super-complexes. Others have found that a range between 2 and 6 g/g [39] or between 4 and 10 g/g [65] does not signifi-cantly change the respiratory chain complex and supercomplex pattern but the optimal digitonin concentration may depend on the tissue source of the sample [14]. We extracted an HEK293T cell pellet derived from a third of a T75 flask in 70–80 µl extraction buffer containing 2% or 3% digitonin and found essentially the same pattern of respira-tory chain enzyme complexes and supercomplexes. However, 1% digitonin was too low to solubilize the respiratory chain enzyme complexes. Like for *n*-dodecyl-β-D-maltoside, we suggest conducting a titration with digitonin to opti-mize the protocol when studying a new cell line or tissue.

To validate our two-dimensional BN/SDS-PAGE system, we investigated a late step of Complex I assembly by transiently knocking down the expression of subunits of the N- and Q-modules. Immunodetection with antibodies against Complex I subunits revealed a pattern of discrete spots on the two-dimensional blot that fitted the migration of constituent subunits of a partly preassembled N-module and a large subassembly consisting of the Q- and P-modules, in agreement with the known assembly pathway [63]. The results substantiate that our two-dimensional BN/SDS-PAGE system yields qualitative infor-mation about the assembly status of OXPHOS complexes, permitting the identification of constituent subunits of assembly intermediates, e.g., in cells from patients with mutations in genes involved in the biosynthesis of OXPHOS complexes [27].

Our BN- and CN-PAGE laboratory protocol is particularly suitable for (1) the characterization of OXPHOS enzyme complexes in samples from patients suspected of OXPHOS deficiency, (2) the study of effects of drugs on the OXPHOS system, and (3) in biomarker studies of OXPHOS function. From patients, often only cultured dermal fibroblasts are avail-able for study. Our protocol is especially adapted for small-scale cell culture samples by using a simplified extraction step to prepare the samples and mini-gels to resolve samples. Complex V is increasingly recognized as a cause of disease [66]. We have improved the sensitivity of the in-gel Complex V staining with a short enhancement step that allows Com-plex V activity measurement in <5 µg of whole-cell extract. We expect that the improved Complex V assay will facilitate the diagnosis of Complex V deficiency significantly.

## Conclusions and limitations

Validation of our recently published step-by-step BN- and CN-PAGE protocol indicates that it is adaptable and yields robust results in downstream western blot analysis and in-gel enzyme activity staining experiments. Our protocol includes a simplified extraction method for small samples and discusses both BN- and CN-PAGE applications. In-gel Complex IV staining was not particularly sensitive for small samples, and we were unable to stain gels for Complex III activity but, by using a simple enhancement step, we were able to increase the sensitivity of in-gel Complex V activity staining markedly.

## Supporting information

**S1 File.** Step-by-step protocol, also available on protocols.io, https://doi.org/10.17504/protocols.io.6qpvrkdrolmk/v1.
(PDF)

**S2 File.** Raw, uncropped images of gels and blots, and data and calculations of Figs 1 and 3–8.
(XLSX)

**S1 Table.** siRNA species.
(PDF)

**S2 Table.** Primary antibodies.
(PDF)

**S3 Table.** Secondary antibodies.
(PDF)

**S1 Fig.** In-gel activity and western blot detection of Complex V after BN-PAGE.
(PDF)

**S2 Fig.** Western blot analysis of a 1% digitonin extract from HEK293T cells.
(PDF)

## Acknowledgments

We would like to acknowledge Dr Sîon L. Williams who developed the original method in our lab.

## Author contributions

**Conceptualization:** Jan-Willem Taanman.

**Data curation:** Jan-Willem Taanman.

**Formal analysis:** Jan-Willem Taanman.

**Funding acquisition:** Jan-Willem Taanman.

**Investigation:** Jana Aref, Seungtae Lee, Supachaya Sriphoosanaphan, Micol Falabella, Shi-Yu Yang, Jan-Willem Taanman.

**Methodology:** Jana Aref, Seungtae Lee, Supachaya Sriphoosanaphan, Micol Falabella, Jan-Willem Taanman.

**Resources:** Supachaya Sriphoosanaphan, Micol Falabella, Jan-Willem Taanman.

**Supervision:** Jan-Willem Taanman.

**Visualization:** Jan-Willem Taanman.

**Writing – original draft:** Jana Aref, Jan-Willem Taanman.

**Writing – review & editing:** Seungtae Lee, Supachaya Sriphoosanaphan, Micol Falabella, Shi-Yu Yang.

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
