## [Decision Letter · Decision Letter 0]

5 Aug 2025

PONE-D-25-37026Validation of blue- and clear-native polyacrylamide gel electrophoresis protocols to characterize mitochondrial oxidative phosphorylation complexesPLOS ONE

Dear Dr. Taanman,

Thank you for submitting your manuscript to PLOS ONE. After careful consideration, we feel that it has merit but does not fully meet PLOS ONE’s publication criteria as it currently stands. Therefore, we invite you to submit a revised version of the manuscript that addresses the points raised during the review process.

We look forward to receiving your revised manuscript.

Kind regards,

Peng Gao, Ph.D.

Academic Editor

PLOS ONE

Journal Requirements:

3. To comply with PLOS One submissions requirements, in your Methods section, please provide additional information regarding the experiments involving animals and ensure you have included details on (1) methods of sacrifice, (2) methods of anesthesia and/or analgesia, and (3) efforts to alleviate suffering.

5. Please note that funding information should not appear in any section or other areas of your manuscript. We will only publish funding information present in the Funding Statement section of the online submission form. Please remove any funding-related text from the manuscript.

6. Please amend your list of authors on the manuscript to ensure that each author is linked to an affiliation. Authors’ affiliations should reflect the institution where the work was done (if authors moved subsequently, you can also list the new affiliation stating “current affiliation:….” as necessary).

7. Your abstract cannot contain citations. Please only include citations in the body text of the manuscript, and ensure that they remain in ascending numerical order on first mention.

8. Please remove all personal information, ensure that the data shared are in accordance with participant consent, and re-upload a fully anonymized data set.

9. We note you have not yet provided a protocols.io PDF version of your protocol and/or a protocols.io DOI. When you submit your revision, please provide a PDF version of your protocol as generated by protocols.io (the file will have the protocols.io logo in the upper right corner of the first page) as a Supporting Information file. The filename should be S1_file.pdf, and you should enter “S1 File” into the Description field. Any additional protocols should be numbered S2, S3, and so on. Please also follow the instructions for Supporting Information captions [https://journals.plos.org/plosone/s/supporting-information#loc-captions]. The title in the caption should read: “Step-by-step protocol, also available on protocols.io.”

Please assign your protocol a protocols.io DOI, if you have not already done so, and include the following line in the Materials and Methods section of your manuscript: “The protocol described in this peer-reviewed article is published on protocols.io (https://dx.doi.org/10.17504/protocols.io.[...]) and is included for printing purposes as S1 File.” You should also supply the DOI in the Protocols.io DOI field of the submission form when you submit your revision.

If you have not yet uploaded your protocol to protocols.io, you are invited to use the platform’s protocol entry service [https://www.protocols.io/we-enter-protocols] for doing so, at no charge. Through this service, the team at protocols.io will enter your protocol for you and format it in a way that takes advantage of the platform’s features. When submitting your protocol to the protocol entry service please include the customer code PLOS2022 in the Note field and indicate that your protocol is associated with a PLOS ONE Lab Protocol Submission. You should also include the title and manuscript number of your PLOS ONE submission.

Reviewers' comments:

Reviewer's Responses to Questions

**Comments to the Author**

1. Does the manuscript report a protocol which is of utility to the research community and adds value to the published literature?

Reviewer #1: Yes

Reviewer #2: Yes

2. Has the protocol been described in sufficient detail?

To answer this question, please click the link to protocols.io in the Materials and Methods section of the manuscript (if a link has been provided) or consult the step-by-step protocol in the Supporting Information files.

The step-by-step protocol should contain sufficient detail for another researcher to be able to reproduce all experiments and analyses.

Reviewer #1: Partly

Reviewer #2: Yes

3. Does the protocol describe a validated method?

Reviewer #1: Yes

Reviewer #2: Yes

4. If the manuscript contains new data, have the authors made this data fully available?

Reviewer #1: Yes

Reviewer #2: Yes

**5. Is the article presented in an intelligible fashion and written in standard English?**

Reviewer #1: Yes

Reviewer #2: Yes

6. Review Comments to the Author

Reviewer #1: The study presents a comprehensive validation of BN-PAGE and CN-PAGE protocols for characterizing OXPHOS complexes, with rigorous experimental design and robust supporting data. The following points should be addressed:

1.The authors should provide an explanation for the observed enhanced Complex IV staining in liver mitochondria relative to cultured cells. Given the liver's well-established high mitochondrial density and elevated metabolic activity, these inherent biological characteristics likely contribute to the differential staining outcomes. Clarification of this relationship would facilitate proper interpretation of the tissue-specific results.

2.For the technical limitations regarding Complex III/IV staining, additional discussion of potential underlying factors (e.g., enzyme stability or detergent compatibility) and attempted optimization strategies would strengthen the methodology section.

3.The wide range of protein loading quantities (5-30 μg) warrants clearer justification, especially whether this reflects biological sample differences or stain-specific sensitivity requirements.

4.The protocol's innovative aspects - particularly the streamlined extraction and enhanced Complex V detection - should be more prominently featured with direct comparisons to conventional methods.

5. Please discuss the similarities, differences, and innovations of this study in comparison to the referenced work (DOI: 10.3791/59294), particularly focusing on methodological advancements and biological implications.

6.Finally, a brief discussion of potential clinical applications, especially for mitochondrial disorder diagnostics, and sample-specific adaptation considerations would enhance the paper's translational impact.

Reviewer #2: This article validates a blue and transparent natural polyacrylamide gel electrophoresis (BN-PAGE and CN-PAGE) protocol for characterizing mitochondrial oxidative phosphorylation complexes. The study clearly describes the experimental protocol and provides sufficient experimental data to support its conclusions. This research has potential practical value as it simplifies the analysis process of mitochondrial oxidative phosphorylation complexes and enhances the sensitivity of certain enzyme activity staining.

Here are two suggestions:

1. The results section can be more concise and clear. For example, the description of the Western blot results can be simplified, with a focus on highlighting the key findings.

2. The discussion section can also explore the application prospects of this protocol in future research. For instance, it can discuss the potential applications of this protocol in diagnosing mitochondrial diseases and studying the effects of drugs on the OXPHOS system.

7. PLOS authors have the option to publish the peer review history of their article (what does this mean? ). If published, this will include your full peer review and any attached files.

**Do you want your identity to be public for this peer review?** For information about this choice, including consent withdrawal, please see our Privacy Policy .

Reviewer #1: No

Reviewer #2: No

---

## [Author Response · Author response to Decision Letter 1]

12 Aug 2025

Dr Peng Gao, Editor ─ PLOS ONE

12 August 2025

Dear Dr Gao,

Re: Validation of blue- and clear-native polyacrylamide gel electrophoresis protocols to characterize mitochondrial oxidative phosphorylation complexes (re-submission)

We thank you for your time and consideration concerning our manuscript. We thank you for sending us the constructive comments from the two reviewers and giving us the opportunity to address their concerns.

We hope that our revised manuscript is now acceptable for publication in PLOS ONE.

As suggested, regarding the experiments involving animals, we have now included details on (1) methods of sacrifice, (2) methods of anesthesia and (3) efforts to alleviate suffering.

Reviewer #1:

The study presents a comprehensive validation of BN-PAGE and CN-PAGE protocols for characterizing OXPHOS complexes, with rigorous experimental design and robust supporting data. The following points should be addressed:

We thank Reviewer 1 for the time and consideration to review our manuscript.

1. The authors should provide an explanation for the observed enhanced Complex IV staining in liver mitochondria relative to cultured cells. Given the liver's well-established high mitochondrial density and elevated metabolic activity, these inherent biological characteristics likely contribute to the differential staining outcomes. Clarification of this relationship would facilitate proper interpretation of the tissue-specific results.

We have now provided an explanation for the observed enhanced Complex IV staining in liver mitochondria relative to cultured cells in the Discussion, as suggested by Reviewer 1.

2. For the technical limitations regarding Complex III/IV staining, additional discussion of potential underlying factors (e.g., enzyme stability or detergent compatibility) and attempted optimization strategies would strengthen the methodology section.

We explain in the text that in-gel Complex IV activity staining is not particularly sensitive and that differences in OXPHOS enzyme content in the liver mitochondrial samples relative to the whole cell extract samples are likely to explain the inability to detect Complex IV activity by in-gel staining. We mention in the text that also others [22,42] have been unable to stain gels with cultured cell fractions for Complex IV activity in a conclusive manner.

To our knowledge, only one paper has claimed to achieve in-gel Complex III activity staining [21]. The authors used a bovine heart mitochondrial sample and incubated the gel in 0.5 mg/ml of diaminobenzidine (DAB) in 50 mM sodium phosphate (pH 7.2). No explanation is given why this incubation should result in a specific Complex III staining. Possibly, a peroxidase comigrating with Complex III oxidizes DAB, producing a brown indamine polymer. We have attempted to stain gels for Complex III with DAB but this did not result in any staining.

3. The wide range of protein loading quantities (5-30 μg) warrants clearer justification, especially whether this reflects biological sample differences or stain-specific sensitivity requirements.

We have now added a paragraph to the Discussion, which discusses protein loading quantities.

4. The protocol's innovative aspects - particularly the streamlined extraction and enhanced Complex V detection - should be more prominently featured with direct comparisons to conventional methods.

We have now mentioned the innovative streamlined extraction and enhanced Complex V detection in the first paragraph of the Discussion.

5. Please discuss the similarities, differences, and innovations of this study in comparison to the referenced work (DOI: 10.3791/59294), particularly focusing on methodological advancements and biological implications.

We have now included the paper by Cuillerier & Burelle (DOI: 10.3791/59294) in our References ([14]) and refer to it in the manuscript twice.

6. Finally, a brief discussion of potential clinical applications, especially for mitochondrial disorder diagnostics, and sample-specific adaptation considerations would enhance the paper's translational impact.

We have now added an extra paragraph at the end of the Discussion to discuss the potential applications of this protocol in diagnosing mitochondrial diseases and studying the effects of drugs on the OXPHOS system.

Reviewer #2: This article validates a blue and transparent natural polyacrylamide gel electrophoresis (BN-PAGE and CN-PAGE) protocol for characterizing mitochondrial oxidative phosphorylation complexes. The study clearly describes the experimental protocol and provides sufficient experimental data to support its conclusions. This research has potential practical value as it simplifies the analysis process of mitochondrial oxidative phosphorylation complexes and enhances the sensitivity of certain enzyme activity staining.

Here are two suggestions:

We thank Reviewer 2 for the time and consideration to review our manuscript.

1. The results section can be more concise and clear. For example, the description of the Western blot results can be simplified, with a focus on highlighting the key findings.

As this is a “Lab Protocol Article”, we feel that a thorough description of the results is needed for those who are new to the technique.

2. The discussion section can also explore the application prospects of this protocol in future research. For instance, it can discuss the potential applications of this protocol in diagnosing mitochondrial diseases and studying the effects of drugs on the OXPHOS system.

We have now added an extra paragraph at the end of the Discussion to discuss the potential applications of this protocol in diagnosing mitochondrial diseases and studying the effects of drugs on the OXPHOS system.

On behalf of all authors.

Yours faithfully,

Jan-Willem Taanman, PhD

Associate Professor of Clinical Neurosciences

---

## [Decision Letter · Decision Letter 1]

26 Aug 2025

Validation of blue- and clear-native polyacrylamide gel electrophoresis protocols to characterize mitochondrial oxidative phosphorylation complexes

PONE-D-25-37026R1

Dear Dr. Taanman,

We’re pleased to inform you that your manuscript has been judged scientifically suitable for publication and will be formally accepted for publication once it meets all outstanding technical requirements.

Kind regards,

Peng Gao, Ph.D.

Academic Editor

PLOS ONE

Additional Editor Comments (optional):

Reviewers' comments:

Reviewer's Responses to Questions

**Comments to the Author**

1. Does the manuscript report a protocol which is of utility to the research community and adds value to the published literature?

Reviewer #1: Yes

Reviewer #2: Yes

2. Has the protocol been described in sufficient detail?

To answer this question, please click the link to protocols.io in the Materials and Methods section of the manuscript (if a link has been provided) or consult the step-by-step protocol in the Supporting Information files.

The step-by-step protocol should contain sufficient detail for another researcher to be able to reproduce all experiments and analyses.

Reviewer #1: Yes

Reviewer #2: Yes

3. Does the protocol describe a validated method?

Reviewer #1: Yes

Reviewer #2: Yes

4. If the manuscript contains new data, have the authors made this data fully available?

Reviewer #1: Yes

Reviewer #2: Yes

**5. Is the article presented in an intelligible fashion and written in standard English?**

Reviewer #1: Yes

Reviewer #2: Yes

6. Review Comments to the Author

Reviewer #1: The authors have fully addressed and supplemented the previous reviewer comments, and conducted in-depth discussions on the key issues raised.

Reviewer #2: This article validates a blue and transparent natural polyacrylamide gel

electrophoresis (BN-PAGE and CN-PAGE) protocol for characterizing mitochondrial

oxidative phosphorylation complexes. The study clearly describes the experimental

protocol and provides sufficient experimental data to support its conclusions. This

research has potential practical value as it simplifies the analysis process of

mitochondrial oxidative phosphorylation complexes and enhances the sensitivity of

certain enzyme activity staining.

7. PLOS authors have the option to publish the peer review history of their article (what does this mean? ). If published, this will include your full peer review and any attached files.

**Do you want your identity to be public for this peer review?** For information about this choice, including consent withdrawal, please see our Privacy Policy .

Reviewer #1: No

Reviewer #2: No

---

## [Editor Report · Acceptance letter]

PONE-D-25-37026R1

PLOS ONE

Dear Dr. Taanman,

I'm pleased to inform you that your manuscript has been deemed suitable for publication in PLOS ONE. Congratulations! Your manuscript is now being handed over to our production team.

Kind regards,

on behalf of

Professor Peng Gao

Academic Editor

PLOS ONE